# Dietary Dityrosine Impairs Glucose Homeostasis by Disrupting Thyroid Hormone Signaling in Pancreatic β-Cells

**DOI:** 10.3390/foods14183220

**Published:** 2025-09-17

**Authors:** Yueting Ge, Boyang Kou, Chunyu Zhang, Chengjia Gu, Lin Cheng, Yonghui Shi, Guowei Le, Wei Xu

**Affiliations:** 1Henan Key Laboratory of Tea Plant Biology, College of Tea and Food Science, Xinyang Normal University, Xinyang 464000, China; ytge@xynu.edu.cn (Y.G.);; 2State Key Laboratory of Food Science and Technology, School of Food Science and Technology, Jiangnan University, Wuxi 214122, China; 3Dabie Mountain Laboratory, Xinyang 464000, China; 4Henan International Joint Laboratory of Tea-Oil Tree Biology and High-Value Utilization, College of Tea and Food Science, Xinyang Normal University, Xinyang 464000, China

**Keywords:** dityrosine, glucose metabolism, thyroid hormone signaling, oxidative stress, inflammation, protein oxidation

## Abstract

Growing evidence links processed red meat consumption to increased diabetes risk, with oxidized proteins and/or amino acids proposed as potential mediators. We investigated whether dityrosine (Dityr), a key oxidation biomarker in high-oxidative pork (HOP) and structural analog of thyroid hormone T3, mediates HOP-induced glucose dysregulation via thyroid hormone (TH) signaling disruption. C57BL/6J mice were fed control, low-oxidative pork (LOP), HOP, LOP + Dityr, or Dityr diets for 12 weeks. HOP and Dityr impaired glucose tolerance and induced hyperglycemia and hypoinsulinemia. Both induced oxidative stress and inflammation that partly contributed to pancreatic β-cell dysfunction and reduction in insulin secretion. Crucially, they downregulated pancreatic thyroid hormone receptor β1 (TRβ1) and monocarboxylate transporter 8 (MCT-8), impairing TH signaling. This reduced TH transport in pancreatic tissue and triggered β-cell apoptosis by modulating TRβ1-mediated expression of TH-responsive genes and proteins involved in pancreatic function, ultimately leading to diminished insulin secretion and elevated blood glucose levels. Dityr alone recapitulated the metabolic and molecular disruptions of HOP. We conclude that Dityr drives HOP-induced glucose metabolism disorders primarily by disrupting TH signaling, along with promoting oxidative stress and inflammation that collectively impair β-cell function. Minimizing dietary Dityr exposure via modified cooking methods or antioxidant-rich diets may mitigate diabetes risk.

## 1. Introduction

Type 2 diabetes (T2D) represents a global public health crisis, characterized by insulin resistance and pancreatic β-cell dysfunction, with dietary factors playing a pivotal role in its pathogenesis [1]. Prospective cohort studies link elevated intake of red and processed meats to a higher risk of T2D [2,3], particularly when prepared using high-temperature methods (e.g., grilling, frying) that generate harmful substances, such as protein oxidation products [4], polycyclic aromatic hydrocarbons [5], and heterocyclic aromatic amines [6]. Although the detrimental effects of red and processed meats can be ascribed to multiple constituents, oxidized proteins and amino acids constitute a significant factor linking their intake to T2D risk [7]. Research indicates that T2D is associated with the pathological accumulation of oxidized proteins [8]. Our previous studies demonstrated that high-temperature and high-pressure (HTP) processing of pork induces high levels of protein oxidation, generating various oxidation products such as dityrosine (Dityr), advanced oxidation protein products (AOPPs), advanced glycation end products (AGEs), and the lipid peroxidation marker malondialdehyde (MDA) [9]. These compounds diminish antioxidant defenses (e.g., glutathione peroxidase [GSH-Px], superoxide dismutase [SOD], total antioxidant capacity [T-AOC]) and induce systemic oxidative stress (through reactive oxygen species [ROS] generation). Among these oxidation products, Dityr is a stable biomarker of protein oxidation formed through tyrosine cross-linking. It accumulates significantly in processed meats and has been implicated in metabolic dysregulation [7,10]. Exogenously ingested Dityr can be absorbed through the intestine, enter systemic circulation, accumulate in internal organs, and exert pathological effects on target organs [7,10]. Dityr also serves as a biomarker for systemic oxidative stress, and its accumulation in the body has been linked to aging and the development of age-related diseases [11,12]. Animal studies have shown that exposure to Dityr impairs pancreatic function in experimental models, leading to glucose metabolism dysregulation, elevated blood glucose levels, and reduced insulin secretion [12,13].

Dityr exhibits high structural similarity to triiodothyronine (T3) [14], a key thyroid hormone critical for cellular growth, development, and metabolism. Thyroid hormones (THs) play essential roles in systemic glucose homeostasis [14], with clinical evidence indicating that low circulating TH levels correlate with increased diabetes risk through direct modulation of pancreatic β-cell function and insulin secretion [15]. THs exert these effects primarily via binding to thyroid hormone receptor β1 (TRβ1) in pancreatic β-cells, a process dependent on cellular uptake by monocarboxylate transporter 8 (MCT-8) [16]. Notably, this structural homology enables Dityr to function as a competitive antagonist of TH signaling by occupying the ligand-binding domain of TRβ1 in pancreatic cells [13]. Consequently, Dityr disrupts TH-mediated transcriptional regulation, specifically impairing T3-dependent insulin synthesis and secretion, contributing to hyperglycemia [13].

Our prior research demonstrated that dietary intake of high-oxidative pork (HOP) induced by HTP cooking impairs pancreatic insulin secretion and induces glucose metabolism disorders in mice. In this process, protein oxidation products serve as a critical mediator [4]. This evidence prompts a critical question regarding the specific component responsible for HOP-mediated impairment of systemic glucose homeostasis. Dityr, recognized both as a biomarker of protein oxidation damage and a structural analog of TH T3, may potentially contribute to HOP-induced glucose dysregulation. Building upon these findings, we therefore hypothesized that Dityr, generated during HTP processing, is the primary mediator of HOP-induced glucose metabolism disorders by disrupting TH signaling. To test this hypothesis, we determined whether purified Dityr recapitulates the effects of HOP on glucose homeostasis, systemic/pancreatic oxidative stress and inflammation, and TH signaling (TRβ1/MCT-8) and downstream β-cell function.

## 2. Materials and Methods

### 2.1. Pork Sample Preparation

Porcine hind leg muscles were acquired from a local slaughterhouse (Wuxi, China), homogenized, and immediately vacuum-packed. Two oxidative treatment groups were prepared in accordance with our earlier reported approach [9]: (1) Low-oxidative pork (LOP) was produced via sous-vide cooking (70 °C for 1 h) of 5-mm thick portions (100 g/vacuum bag), while (2) HOP underwent HTP processing (121 °C, 0.2 MPa for 1 h) with identical portioning. Following cooking, samples were homogenized, lyophilized, and preserved at −80 °C in vacuum-sealed packages until dietary formulation. The oxidative profiles of differentially cooked pork were quantitatively characterized in our prior investigation [9].

### 2.2. Animals and Experimental Design

All animal experiments were performed in adherence to the National Guidelines for Experimental Animal Welfare and received approval from the Animal Care and Use Ethics Committee of Jiangnan University (JN. No20190330c0900820[47]). Fifty 4-week-old male specific pathogen-free (SPF) C57BL/6J mice (weighing 17–20 g) were sourced from the Model Animal Research Center of Nanjing University (License No. SCXK (Su) 2015-0001) and maintained in the animal facility at Jiangnan University under SPF conditions. Environmental conditions were controlled at 22 ± 2 °C, 50 ± 10% relative humidity, with a 12-h light/12-h dark cycle (illumination from 08:00 to 20:00). All mice had ad libitum access to food and water for the duration of the study.

After a 7-day acclimation period, mice were randomly divided into five groups (*n* = 10 per group) as follows: (1) Control (CON): soy protein isolate-based diet; (2) LOP: diet containing low-oxidation pork; (3) HOP: diet containing high-oxidation pork; (4) LOP + Dityr: LOP diet supplemented with Dityr (amount equivalent to the difference between LOP and HOP); (5) Dityr: soy protein isolate diet supplemented with Dityr (amount matching HOP levels). All diets adhered to AIN-93G guidelines (complete ingredient list in Appendix A), with protein content standardized at 20% across experimental groups. LOP and HOP diets varied solely by pork preparation method, while HOP, LOP + Dityr, and Dityr groups maintained identical Dityr concentrations.

Total Dityr content in LOP and HOP, as determined by a previously described method [9], is presented in Appendix A. Dietary fat levels were standardized according to measured values (Appendix A). Body weight and food consumption were monitored on a weekly basis throughout the 12-week experimental period.

### 2.3. Oral Glucose Tolerance Test (OGTT)

An OGTT was conducted at week 12 by gavage of 2 g/kg glucose after fasting overnight at week 12. Tail vein blood samples were obtained at 0, 15, 30, 60, 90, and 120 min following the glucose administration. Blood glucose levels were measured using a One Touch Ultra glucometer (LifeScan, Inc., Malvern, PA, USA), and the corresponding area under the curve (AUC) within 120 min was computed.

### 2.4. Tissue Sample Collection

After 12 weeks of feeding, all mice underwent a 12-h (water ad libitum) and were anesthetized by an intraperitoneal injection of sodium pentobarbital. Blood samples were drawn via orbital puncture, followed by cervical dislocation. Subsequent to blood collection, glucose and ROS levels were immediately assessed. The remaining blood was transferred into heparinized tubes, kept at 4 °C for 30 min, and subsequently centrifuged (3500× *g*, 15 min, 4 °C) to obtain plasma. Pancreatic tissues were rapidly excised, and portions were weighed to prepare 10% (*w*/*v*) tissue homogenates in PBS for immediate ROS determination. Remaining homogenates were centrifuged (3500× *g*, 10 min, 4 °C), and supernatants preserved at −80 °C for subsequent biochemical analyses. Pancreatic tail sections were fixed in 4% formaldehyde solution at 4 °C for histopathological examination. For RNA extraction, 50 mg pancreatic tissue samples were immediately placed in RNase-free tubes containing 600 μL TRIzol reagent and frozen at −80 °C. Residual pancreatic tissues were rapidly frozen in liquid nitrogen and maintained at −80 °C for subsequent analysis.

### 2.5. Determination of Oxidative Damage and Oxidative Stress Status in Mice

Protein oxidative damage was evaluated by measuring the levels of Dityr, AOPPs and AGEs in plasma and pancreas tissue using commercial ELISA kits (Dityr: Cat. EHJ-96857m; AOPPs: Cat. EHJ-96098m; AGEs: Cat. EHJ-30217m; Xiamen Huijia Bioengineering Institute, Xiamen, China) in accordance with the manufacturer’s instructions. Lipid peroxidation was evaluated by measuring MDA levels in plasma and pancreas tissue with a commercial thiobarbituric acid reactive substances (TBARS) assay kit (Cat. A003-1; Nanjing Jiancheng Bioengineering Institute, Nanjing, China). Oxidative stress status was assessed by evaluating key markers: T-AOC, SOD, GSH-Px, and ROS. ROS levels in blood and the pancreas were quantified by employing a luminol-dependent chemiluminescence assay based on a previously described method [17], with results expressed as relative light units (RLUs). T-AOC along with SOD and GSH-Px activities in both compartments were analyzed using commercial kits (T-AOC: Cat. A015-1; SOD: Cat. A001-1; GSH-Px: Cat. A005-1; Nanjing Jiancheng Bioengineering Institute, China) following the manufacturer’s guidelines.

### 2.6. Analysis of Inflammatory Cytokines and Metabolic Hormones

Pro-inflammatory cytokines (tumor necrosis factor-alpha [TNF-α]: Cat. EHJ-45111m; interleukin-1 beta [IL-1β]: EHJ-30568m; IL-6: Cat. EHJ-95903m) and the anti-inflammatory cytokine IL-10 (Cat. EHJ-47391m) in pancreatic tissue and plasma, along with plasma lipopolysaccharide (LPS: Cat. EHJ-95884m) and LPS-binding protein (LBP: Cat. EHJ-97482m), were quantified with commercial ELISA kits (Xiamen Huijia Bioengineering Institute, Xiamen, China) following the manufacturer’s instructions. Concurrently, fasting plasma insulin (Cat. EHJ-30401m) and glucagon (Cat. EHJ-96109m) concentrations were assessed using ELISA kits obtained from the same supplier. All samples were measured in triplicate, and absorbance readings were taken at 450 nm with a microplate reader (BioTek Instruments, Winooski, VT, USA).

### 2.7. Histological Analysis of the Pancreas

Pancreatic tissues samples were immersed in 4% paraformaldehyde and fixed for 24-48 h at room temperature. Subsequent standard processing included dehydration through a graded ethanol series, clearing in xylene, and embedding in paraffin. Sections were then sliced to a 5-μm thickness employing a microtome (Leica, Wetzlar, Germany). Subsequently, tissue sections underwent deparaffinization, rehydration, and hematoxylin and eosin (H&E, Baton Rouge, LA, USA) staining in line with standard protocols. The images were captured with a CX31 RTSF microscope (Olympus Corporation, Tokyo, Japan) under 200× magnification. Pancreatic islet areas were quantified utilizing Image-Pro Plus software (version 6.0; Media Cybernetics, Inc., Rockville, MD, USA).

### 2.8. Gene Expression Analysis by RT qPCR

Total RNA from pancreatic tissue was isolated with TRIzol reagent. RNA concentration and purity were assessed with a NanoDrop 1000 spectrophotometer (Thermo Fisher Scientific, Waltham, MA, USA), while its integrity was confirmed by agarose gel electrophoresis. Isolated RNA was subsequently reverse-transcribed to cDNA with HiScript II Q RT SuperMix for qPCR (Vazyme Biotech, Nanjing, China), per the manufacturer’s protocol. Gene expression levels were quantified employing an ABI 7900 HT Fast Real-Time PCR System (Applied Biosystems, Foster City, CA, USA). Each 10 µL qPCR reaction mixture comprised 1 µL of cDNA, 0.4 µL each of forward and reverse primers (10 µM; sequences provided in Appendix A), 5 µL of 2× ChamQ Universal SYBR qPCR Master Mix (Vazyme Biotech, Cat Q711), and 3.2 µL of nuclease-free water. The amplification conditions were: 95 °C for 5 min (initial denaturation); followed by 40 cycles of 95 °C for 20 s (denaturation), 60 °C for 30 s (annealing), and 72 °C for 1 min (extension); then 72 °C for 2 min (final extension). A melt curve analysis was performed immediately after the amplification cycles to verify the specificity of the PCR products. The reaction mixture was then held at 4 °C. Gene expression levels were normalized against β-actin, and relative quantification was performed via the 2−ΔΔCt method [18]. All primers were designed and commercially synthesized by GENEWIZ (Suzhou, China).

### 2.9. Western Blot Analysis

Western blotting was performed following a previously established method [19], with minor modifications. Pancreatic proteins were extracted with RIPA lysis buffer supplemented with phenylmethylsulfonyl fluoride (PMSF) protease inhibitor (Beyotime Biotechnology, Shanghai, China). Protein concentrations were measured via the bicinchoninic acid (BCA) assay kit (Pierce, Rockford, IL, USA) in accordance with the manufacturer’s protocol. The protein extracts were resolved on 10% SDS-polyacrylamide gels after adjusted to the same concentration and transferred to the nitrocellulose membranes (Bio-Rad, Hercules, CA, USA). The membranes were incubated overnight at 4 °C with primary antibodies diluted in 5% non-fat milk prepared in Tris-buffered saline containing Tween 20 (TBST). The antibodies used were: anti-TRβ1 (1:1000; Abcam, Cambridge, UK, Cat. ab180612), anti-B-cell lymphoma 2 (Bcl-2, 1:1000; Abcam, Cat. ab182858,), anti-BCL2-associated X protein (Bax, 1:1000; Abcam, Cat. ab32503), anti-cysteinyl aspartate-specific proteinase 3 (Caspase-3, 1:1000; Abcam, Cat. ab184787), and anti-β-actin (1:1000; Sigma-Aldrich, St. Louis, MO, USA, Cat. A5441). Following three washes (10 min each) with TBST, the membranes were incubated for 1 h at room temperature with a horseradish peroxidase (HRP)-conjugated goat anti-rabbit IgG secondary antibody (1:10,000; Cell Signaling Technology, Danvers, MA, USA, Cat. 7074) diluted in 5% non-fat milk/TBST. Protein bands were detected and captured with an Odyssey infrared imaging system (LI-COR Biosciences, Bourne, MA, USA). Band intensities were quantified with ImageJ software (Version 1.53c; National Institutes of Health, USA). The expression levels of target proteins were normalized to β-actin.

### 2.10. Statistical Analysis

All statistical analyses were performed with SPSS 20.0 (IBM, USA) and the results are presented as mean ± standard error of the mean (SEM). The Shapiro–Wilk test was applied to assess data normality, while Levene’s test was used to verify homogeneity of variances. For datasets violating the normality assumption, appropriate transformations (e.g., logarithmic) were applied. Intergroup differences were evaluated using one-way analysis of variance (ANOVA). When both assumptions (normality and homogeneity of variances) were satisfied, Tukey’s honestly significant difference (HSD) post hoc test was employed for multiple comparisons following a significant ANOVA result. For data where the assumption of homogeneity of variances was violated, the Tamhane’s T2 post hoc test was employed. A probability value of *p* < 0.05 was considered statistically significant. Bar and line graphs were created with GraphPad Prism 8.0 (GraphPad Software, USA). Across all figures and tables, differing superscript letters denote significant differences (*p* < 0.05) between groups.

## 3. Results

### 3.1. Dityr Increases Body Weight and Fasting Blood Glucose, Decreased Fasting Plasma Insulin, and Impaired Glucose Tolerance

Initial body weights did not differ significantly among the groups (Figure 1A, *p* > 0.05). Following the 12-week intervention, mice in the HOP and LOP + Dityr groups showed significantly greater final body weight (Figure 1B) and weight gain (Figure 1C) compared to those in the CON and LOP groups (*p* < 0.05). The Dityr group also exhibited a marked elevation in final body weight (Figure 1B) and weight gain of mice (Figure 1C) compared with the CON group (*p* < 0.05). Notably, the body weights of the LOP + Dityr and Dityr groups mice were closer to those in the HOP group. Compared with the CON and LOP groups, the HOP, LOP + Dityr, and Dityr groups mice exhibited significantly higher fasting blood glucose (Figure 1D, *p* < 0.05) but significantly lower fasting plasma insulin (Figure 1E, *p* < 0.05). OGTT results (Figure 1H) revealed that blood glucose levels of mice within 120 min after glucose gavage were significantly elevated (*p* < 0.05) in HOP, LOP + Dityr, and Dityr groups, with a corresponding significant increase (*p* < 0.05) in AUC values (Figure 1I). No significant intergroup differences were observed in fasting plasma glucagon (Figure 1F) and food intake (Figure 1G, *p* > 0.05).

### 3.2. Dityr Induces Systemic and Pancreatic Oxidative Damage

As shown in Table 1, plasma levels of Dityr, AOPPs, and MDA were significantly increased in HOP, LOP + Dityr, and Dityr groups relative to the CON and LOP groups (*p* < 0.05), along with significantly increased plasma AGEs levels in the HOP group (*p* < 0.05). Pancreatic Dityr content was markedly higher in the HOP, LOP + Dityr, and Dityr groups versus CON and LOP groups (*p* < 0.05). Notably, pancreatic AOPPs and AGEs levels were significantly increased in both HOP and LOP + Dityr groups (*p* < 0.05), while pancreatic MDA levels showed significant elevation in HOP and Dityr groups (*p* < 0.05). Additionally, the Dityr group exhibited markedly increased levels of pancreatic AOPPs and AGEs relative to the CON group (*p* < 0.05).

### 3.3. Dityr Disrupts Systemic and Pancreatic Redox Homeostasis

Table 2 results demonstrated that relative to the CON and LOP groups, the HOP, LOP + Dityr and Dityr groups exhibited markedly elevated plasma ROS levels with markedly decreased GSH-Px activity and T-AOC (*p* < 0.05), along with reduced plasma SOD activity specifically in HOP and Dityr groups. Similarly in pancreatic tissue, these treatment groups showed significantly increased ROS levels accompanied by diminished SOD activity and T-AOC (*p* < 0.05), while the HOP and LOP + Dityr groups additionally displayed reduced GSH-Px activity. Notably, the Dityr group alone demonstrated significantly lower pancreatic GSH-Px activity relative to the CON group (*p* < 0.05). Furthermore, the mRNA expression of genes involved in the antioxidant defense pathway was assessed in pancreatic tissue (Figure 2A). Relative to the CON and LOP groups, the HOP, LOP + Dityr, and Dityr groups exhibited significant downregulation in mRNA expression of NAD(P)H quinone oxidoreductase 1 (NQO-1) and nuclear factor erythroid 2-related factor 2 (Nrf2) (*p* < 0.05). Notably, heme oxygenase-1 (HO-1) mRNA levels in the pancreas were markedly lower in all experimental groups (LOP, HOP, LOP + Dityr, and Dityr) than in the CON group (*p* < 0.05).

### 3.4. Dityr Triggers Systemic and Pancreatic Inflammation Response

Plasma inflammatory markers are presented in Figure 2B–G, showing that compared with CON and LOP groups, HOP, LOP + Dityr, and Dityr groups exhibited significantly elevated plasma LPS (Figure 2B) and TNF-α (Figure 2D) levels coupled with decreased IL-10 (Figure 2G; *p* < 0.05), while LBP levels (Figure 2C) were specifically elevated in the HOP group. When compared solely with the CON group, HOP, LOP + Dityr, and Dityr additionally demonstrated higher plasma IL-6 (Figure 2E) and IL-1β (Figure 2F) levels, with LOP + Dityr and Dityr showing increased LBP (*p* < 0.05). Pancreatic inflammatory profiles (Figure 3) revealed that relative to both CON and LOP groups, HOP, LOP + Dityr, and Dityr displayed elevated IL-1β (Figure 3C) but reduced IL-10 levels (Figure 3D), while IL-6 (Figure 3B) was specifically increased in HOP and LOP + Dityr groups (*p* < 0.05). Compared to the CON group alone, HOP and Dityr showed higher pancreatic TNF-α (Figure 3A), with the Dityr group exhibiting additional IL-6 elevation (*p* < 0.05). Gene expression analysis of pancreatic inflammatory pathways (Figure 3E) demonstrated upregulated IL-1β, TNF-α, and Toll-like receptor 4 (TLR4) mRNA in the HOP, LOP + Dityr, and Dityr groups compared to the CON and LOP groups, along with increased nuclear factor kappa beta (NF-κB) in HOP and LOP + Dityr and decreased IL-10 in these same groups (*p* < 0.05). Furthermore, relative to the CON group, the HOP, LOP + Dityr and Dityr groups showed significantly upregulated mRNA expression of pancreatic myeloid differentiation factor 88 (MyD88), along with elevated IL-6 expression in all treatment groups (including LOP). Notably, the Dityr group specifically demonstrated increased NF-κB and decreased IL-10 mRNA levels (*p* < 0.05).

### 3.5. Dityr Downregulates Pancreatic TRβ1 and MCT-8 to Impair Thyroid Hormone Signaling

To investigate the potential mechanisms underlying Dityr-induced hyperglycemia, we analyzed the gene (Figure 4A) and protein (Figure 4C,D) expression of thyroid hormone receptor TRβ1, along with the gene expression of thyroid hormone transporter MCT-8 (Figure 4B) in pancreatic tissue. Compared with the CON and LOP groups, the HOP, LOP + Dityr, and Dityr groups exhibited significant downregulation of both TRβ1 (at mRNA and protein levels) and MCT-8 (mRNA level) expression in the pancreas (*p* < 0.05). These findings suggest that Dityr may impair thyroid hormone transport in pancreatic tissue and potentially attenuate TRβ1-mediated signaling transduction.

### 3.6. Dityr Exacerbates Pancreatic β-Cell Dysfunction Through Thyroid Hormone-Mediated Apoptosis and Impaired Insulin Secretion

We next examined thyroid hormone-regulated genes and proteins associated with pancreatic β cells apoptosis (Figure 5A) and insulin secretion (Figure 5B,E,F), along with pancreatic histopathological analysis (Figure 5C,D). Relative to the CON and LOP groups, the HOP, LOP + Dityr, and Dityr groups showed significantly downregulated mRNA expression of pancreatic v-maf musculoaponeurotic fibrosarcoma oncogene homologue A (MafA). In contrast, the expression of pancreatic duodenal homeobox-1 (PDX-1) was specifically reduced only in the HOP group (*p* < 0.05). Relative to the CON group alone, LOP + Dityr and Dityr exhibited decreased PDX-1 expression, HOP, LOP + Dityr, and Dityr displayed lowered glucose transporter 2 (GLUT2) expression, and the Dityr group showed downregulated glucokinase (GCK) expression (*p* < 0.05), collectively indicating impaired insulin secretion following Dityr intake. Furthermore, versus CON and LOP groups, HOP and LOP + Dityr groups demonstrated upregulated Bax (mRNA and protein), HOP and Dityr groups showed elevated Caspase-3 mRNA, and all three treatment groups (HOP, LOP + Dityr, and Dityr) exhibited increased Caspase-3 protein alongside downregulated Bcl2 (mRNA and protein) (*p* < 0.05). Compared specifically with the CON group, Dityr displayed higher Bax (mRNA and protein) while LOP + Dityr group showed increased Caspase-3 mRNA (*p* < 0.05), suggesting Dityr induced β-cell apoptosis. Histopathological examination confirmed these findings, revealing significantly reduced islet area ratios in HOP, LOP + Dityr, and Dityr groups, indicative of diminished β-cell mass. These results demonstrate that Dityr plays a crucial role in HOP-induced metabolic dysregulation, potentially through TRβ1-mediated modulation of thyroid hormone target genes/proteins involved in pancreatic function. This mechanism likely promotes β-cell apoptosis, thereby impairing insulin secretion and contributing to systemic hyperglycemia. A summary of the major effects across all dietary groups is provided in Appendix A.

## 4. Discussion

Growing evidence links processed red meat consumption to increased diabetes risk [2], with oxidized proteins and/or amino acids such as Dityr proposed as potential mediators. Our results substantiate this hypothesis by demonstrating that Dityr induces glucose metabolism disorders through thyroid hormone signaling disruption and systemic oxidative stress/inflammation, ultimately leading to pancreatic β-cell dysfunction, impaired insulin secretion, and hyperglycemia in mice. These findings position dietary protein oxidation products as critical drivers of processed meat-associated metabolic dysregulation.

Dityr induced hyperglycemia, hypoinsulinemia, and impaired glucose tolerance in mice. Our results demonstrated that HOP-fed mice exhibited significantly elevated fasting blood glucose and AUC values, along with markedly reduced fasting plasma insulin levels, indicating that HOP consumption impaired glucose tolerance and induced hyperglycemia and hypoinsulinemia, thereby increasing diabetes risk. These findings are consistent with our previous studies [4]. In line with the glucose metabolism dysfunction caused by HOP, Dityr supplementation produced comparable effects, including elevated fasting blood glucose, impaired glucose tolerance, and reduced insulin secretion. Prior studies have similarly reported that Dityr intake increases blood glucose levels while decreasing plasma insulin, which corroborates our current findings [12,13]. Furthermore, HOP-fed mice showed significant body weight gain, again consistent with our earlier observations [4]. Dityr supplementation recapitulated these weight-increasing effects.

Dityr increased the accumulation of oxidative damage markers in mice, attenuated the antioxidant defense capacity, and induced oxidative stress. Dietary oxidation induces systemic oxidative stress [7]. Dietary oxidation products can be absorbed through the intestine and accumulate in circulation, thereby inducing ROS generation and compromising the antioxidant defense capacity of the body. ROS are natural byproducts of cellular metabolism that are normally maintained in dynamic equilibrium through elimination by endogenous antioxidant systems. Excessive ROS production overwhelming antioxidant defenses induces redox imbalance and oxidative stress [7]. The Nrf2-ARE pathway is pivotal for oxidative stress defense [20]. Nrf2, a master transcriptional regulator, upregulates cytoprotective genes (e.g., NQO1, HO-1) upon activation [21]. Impaired activation or deficiency of Nrf2 may disrupt cellular redox homeostasis, resulting in oxidative stress. Notably, Nrf2-knockout cells and tissues exhibit increased ROS production [20]. Our results demonstrated that HOP consumption significantly increased the accumulation of protein oxidation markers (Dityr, AOPPs, and AGEs) and lipid peroxidation products (MDA) in both plasma and pancreatic tissue, while elevating ROS in blood and pancreas. Concurrently, HOP intake markedly reduced T-AOC and antioxidant enzyme activities (SOD and GSH-Px) in plasma and the pancreas, indicating HOP-induced systemic oxidative stress. Furthermore, HOP exposure downregulated mRNA expression for key antioxidant defense-related genes (Nrf2, HO-1, and NQO-1) in pancreatic tissue, which not only corroborates these findings but also suggests that HOP induces oxidative stress by inhibiting the Nrf2-ARE signaling pathway. These findings align with our prior reports [4,9]. Notably, Dityr supplementation significantly increased levels of Dityr, AOPPs, and MDA in plasma, along with Dityr, AOPPs, MDA, and AGEs in pancreatic tissue, while increasing ROS levels in both blood and pancreas. Concurrently, it markedly reduced T-AOC and activities of GSH-Px and SOD in plasma and the pancreas, and suppressed the pancreatic Nrf2/ARE antioxidant pathway. These results demonstrate that Dityr intake promotes accumulation of oxidative damage markers, impairs antioxidant defenses, and induces systemic oxidative stress, with its oxidative damage effects approaching those observed in HOP-fed mice.

Dityr caused a systemic inflammatory response. Our previous study demonstrated that HOP consumption significantly elevated the relative abundance of Escherichia coli in mice colonic contents. As a Gram-negative bacterium, Escherichia coli releases endotoxin LPS, which compromises intestinal barrier integrity and translocates into circulation, thereby inducing systemic inflammation responses [22]. Circulating LPS forms a complex with its binding protein LBP and receptor CD14 (LPS-LBP-CD14), which is subsequently recognized by and binds to TLR4 for intracellular signal transduction [22]. Intracellularly, this signal is propagated through either MyD88-dependent or -independent pathways, ultimately activating NF-κB and other downstream pathways to stimulate the secretion of pro-inflammatory cytokines such as TNF-α, IL-1, and IL-6, thereby promoting systemic inflammation [23]. Our current findings revealed that HOP intake notably raised plasma concentrations of LPS and LBP, increased levels of pro-inflammatory cytokines (IL-1β, IL-6, and TNF-α) in plasma and pancreatic tissue, while markedly reducing anti-inflammatory cytokine IL-10 levels, confirming HOP-induced systemic inflammation. Furthermore, HOP consumption significantly upregulated pancreatic mRNA expression of inflammation-related genes (TLR4, MyD88, NF-κB, TNF-α, IL-6, and IL-1β) and downregulated IL-10 expression, demonstrating that HOP likely induces inflammatory responses through activation of the TLR4/MyD88/NF-κB signaling pathway. These findings align with our previous research [4,9]. Notably, Dityr supplementation similarly increased plasma LPS and LBP levels, elevated TNF-α, IL-6 and IL-1β in plasma and pancreas, activated the pancreatic TLR4/MyD88/NF-κB pathway, and decreased IL-10 levels and gene expression. These effects demonstrate that Dityr can induce comparable inflammatory responses, with severity approaching that of HOP-fed mice. Together, these results establish Dityr as a critical mediator of HOP-induced oxidative stress and inflammation—a conclusion supported by existing literature documenting the pro-oxidative and pro-inflammatory effects of Dityr in experimental models [24,25]. Accumulating evidence indicates that oxidative stress and inflammation accelerate diabetes progression by directly damaging pancreatic tissue and impairing β-cell function [26,27], which may represent a mechanism underlying Dityr-mediated hypoinsulinemia.

Dityr induced apoptosis in pancreatic β-cells and diminished insulin secretion by disrupting thyroid hormone signaling. Insulin, the sole hypoglycemic hormone in mammals, is secreted from pancreatic β-cells and is crucial for regulating systemic glucose homeostasis. The dysfunction of β-cells, typified by a decrease in β-cell mass and impaired insulin secretion, is intimately linked to the pathogenesis and progression of diabetes mellitus [28]. Thyroid hormones are crucial for maintaining glucose homeostasis, regulating pancreatic insulin secretion, along with β-cell proliferation and apoptosis through TRβ1-mediated mechanisms [29]. Studies demonstrate that T3 administration can ameliorate impaired insulin secretion in growth-retarded mice [30]. MafA, a critical transcriptional factor in insulin secretion, regulates genes related to both insulin synthesis and secretion [31]. As a direct target of thyroid hormones in β-cells, MafA enhances glucose-stimulated insulin secretion in neonatal rat islets [32]. Mice lacking MafA display deficient insulin secretion and glucose intolerance postnatally [33]. Thyroid hormones promote the development and functional maturation of β-cell by upregulating MafA expression, thereby improving glucose-stimulated insulin secretion (GSIS)—a process mediated by thyroid hormone receptors (TRs) [34]. Notably, T3 directly binds to the MafA promoter to enhance its transcription [34]. In adult islets, TRβ serves as the predominant TR subtype [34], and T3 treatment upregulates TR expression in insulin-secreting cells [35]. PDX-1 serves as another crucial transcription factor facilitating the development and maturation of pancreatic β-cell [36]. In mature β-cells, PDX-1 maintains normal cellular function and regulates insulin secretion by transcriptionally regulating the insulin gene and genes associated with insulin secretion such as GCK and GLUT2. Impaired insulin secretion is associated with reduced PDX-1 expression [36]. GLUT2 and GCK serve as key mediators of glucose transport and phosphorylation, respectively, thereby governing glucose-sensing processes in pancreatic β-cells [37]. Notably, T3 treatment has been shown to upregulate pancreatic expression of PDX-1, GCK, and GLUT2. Our results revealed that HOP consumption significantly downregulated pancreatic expression of thyroid hormone-regulated insulin synthesis/secretion-related genes (MafA, PDX-1, and GLUT2), TRβ1 (both mRNA and protein), and thyroid hormone transporter MCT-8 mRNA in mice. Notably, Dityr supplementation similarly suppressed pancreatic mRNA expression of MafA, PDX-1, GLUT2, and GCK, while reducing both transcriptional and translational levels of TRβ1 and decreasing MCT-8 expression. These molecular alterations mechanistically support the observed hypoinsulinemia in HOP- and Dityr-treated groups, and are consistent with previous reports demonstrating Dityr-induced reductions in pancreatic TRβ1 and MCT-8 expression [12]. Apoptosis is a critical factor contributing to β-cell dysfunction []. Thyroid hormones play diverse biological roles in regulating cell growth, differentiation, and metabolism. Studies demonstrate that T3 inhibits pancreatic β-cell apoptosis and promotes their growth and proliferation by modulating the expression of the anti-apoptotic protein Bcl-2 and the pro-apoptotic proteins Caspase-3 and Bax, thereby counteracting diabetes induced by streptozotocin and hydrogen peroxide [35,38]. These protective effects are mediated specifically by TRβ1 [39]. Bcl-2, a member of the Bcl-2 protein family, exerts anti-apoptotic effects by localizing to sites of ROS generation, including the endoplasmic reticulum, mitochondrial, and nuclear membranes. Its overexpression suppresses lipid peroxidation and ROS production [40]. In contrast, Bax, another Bcl-2 family member, promotes apoptosis by triggering the release of pro-apoptotic factors and antagonizing the protective function of Bcl-2 [41]. Under physiological conditions, Bcl-2 and Bax maintain relatively stable expression levels and can form heterodimers [41]. Studies indicate that the ratio of these functionally antagonistic proteins is a critical determinant of cell survival [41,42]. Elevated Bax expression promotes the formation of Bax/Bax homodimers, initiating apoptotic signaling. Conversely, high Bcl-2 expression shifts the equilibrium by dissociating Bax/Bax homodimers and favoring the formation of more stable Bcl-2/Bax heterodimers, thereby suppressing apoptosis [41]. Caspase-3, a key executor of apoptosis, plays a pivotal role in modulating intracellular apoptotic signaling pathways, with its expression level reflecting the extent of apoptotic signaling [43,44]. Apoptotic progression is accompanied by increased expression of pro-apoptotic factors such as Bax, Caspase-3, along with decreased levels of anti-apoptotic factors like Bcl-2. Notably, studies demonstrate that T3 suppresses pancreatic β-cell apoptosis by upregulating the anti-apoptotic protein Bcl-2 and simultaneously reducing the expression of pro-apoptotic mediators Caspase-3 and Bax [35,38]. Our findings demonstrated that HOP consumption markedly elevated the mRNA and protein expression of pro-apoptotic factors (Caspase-3 and Bax) and concurrently suppressed anti-apoptotic Bcl-2 expression in pancreatic tissue, indicating that HOP induces β-cell apoptosis through coordinated activation of pro-apoptotic pathways and suppression of anti-apoptotic defenses, ultimately impairing insulin secretion. Histopathological analysis confirmed a significant decrease in β-cell mass in HOP-fed mice, corroborating these molecular observations. Notably, Dityr supplementation similarly triggered pancreatic β-cell apoptosis by elevating Caspase-3 and Bax expression and suppressing Bcl-2, leading to in diminished insulin secretion. The significantly decreased β-cell mass in Dityr-fed mice further validated this conclusion. As a structural analog of thyroid hormone T3, Dityr has been reported to share identical binding sites with T3 on TRβ1. This competitive binding may downregulate TRβ1-mediated expression of downstream genes, thereby attenuating the regulatory effects of T3 on pancreatic insulin synthesis [13]. In the present study, Dityr likely interferes with the endocrine actions of T3 by competing for TRβ1 binding in pancreatic cells and attenuating transcriptional regulation of target genes, contributing at least partially to the observed reduction in insulin secretion.

The detrimental impact of Dityr on pancreatic β-cell function places it among a growing list of dietary and environmental stressors implicated in the pathogenesis of diabetes. It is instructive to compare its mechanisms and effects with other well-established disruptors. For instance, AGEs—which were also significantly elevated in the HOP group in our study (Table 1)—are known to impair β-cell function through receptor (RAGE)-mediated activation of oxidative stress and inflammatory pathways, mechanisms that parallel our observations with Dityr [45]. However, a key distinction lies in the unique action of Dityr as a structural analog of T3, enabling it to serve as a competitive antagonist of thyroid hormone receptor signaling. This mechanism is more akin to certain endocrine-disrupting chemicals (EDCs) that interfere with nuclear hormone receptor function [46], highlighting a novel pathway for dietary-derived β-cell toxicity. Similarly, lipotoxicity, which arises from chronic exposure to high levels of free fatty acids, triggers β-cell apoptosis and dysfunction primarily through endoplasmic reticulum stress and mitochondrial impairment [47]. It is plausible that Dityr-induced oxidative stress could synergize with these pathways, exacerbating metabolic dysfunction. Therefore, while Dityr converges on common downstream endpoints of β-cell failure (oxidative stress, inflammation, apoptosis) shared with AGEs, lipotoxicity, and EDCs, its initiation via the disruption of TH signaling represents a distinct and previously underappreciated dietary mechanism. Our findings underscore the multifactorial nature of diet-induced diabetes and emphasize the importance of developing comprehensive dietary strategies that mitigate exposure to this spectrum of β-cell toxicants, including oxidized amino acids like Dityr.

Beyond elucidating the mechanism, our study underscores the potential of Dityr to serve as a sensitive biomarker for assessing the nutritional risk associated with processed and oxidized protein-rich foods. Quantifying Dityr levels in food products could provide a valuable index of protein oxidation damage and help consumers make informed choices. Furthermore, monitoring circulating or urinary Dityr levels might be explored as a non-invasive strategy to assess individual exposure and associated metabolic health risks. From an interventional perspective, our findings suggest two pragmatic dietary approaches to mitigate Dityr-related risks. First, adopting gentle cooking methods, such as sous-vide or steaming, instead of high-temperature grilling or frying, could significantly reduce the formation of Dityr and other oxidation products in meat [10]. Second, increasing the intake of foods abundant in antioxidants (such as fruits, vegetables, and tea) might counteract the oxidative stress and inflammation triggered by dietary Dityr, potentially attenuating its adverse effects on glucose homeostasis. Future investigation is warranted to confirm the efficacy of these interventions in populations with high consumption of processed meats.

## 5. Conclusions

This study establishes that the protein oxidation biomarker Dityr plays a pivotal role in HOP-induced glucose metabolism disorders. Dityr administration significantly impaired glucose tolerance and induced hyperglycemia and hypoinsulinemia in mice. Mechanistically, Dityr reduced thyroid hormone transport in pancreatic tissue and triggered β-cell apoptosis by modulating TRβ1-mediated expression of thyroid hormone-responsive genes and proteins involved in pancreatic function, ultimately leading to diminished insulin secretion and elevated blood glucose levels. Furthermore, Dityr increased accumulation of oxidative damage markers while reducing systemic antioxidant defenses, resulting in oxidative stress and inflammatory responses that likely contributed to pancreatic β-cell dysfunction and the observed reduction in insulin secretion. These findings suggest that minimizing dietary exposure to protein oxidation products like Dityr through modified cooking methods (e.g., low-temperature processing) or concurrent consumption of antioxidant-rich foods may help mitigate HOP-associated metabolic dysregulation. Such dietary interventions could be particularly relevant for populations at risk of β-cell dysfunction.

## Figures and Tables

**Figure 1 foods-14-03220-f001:**
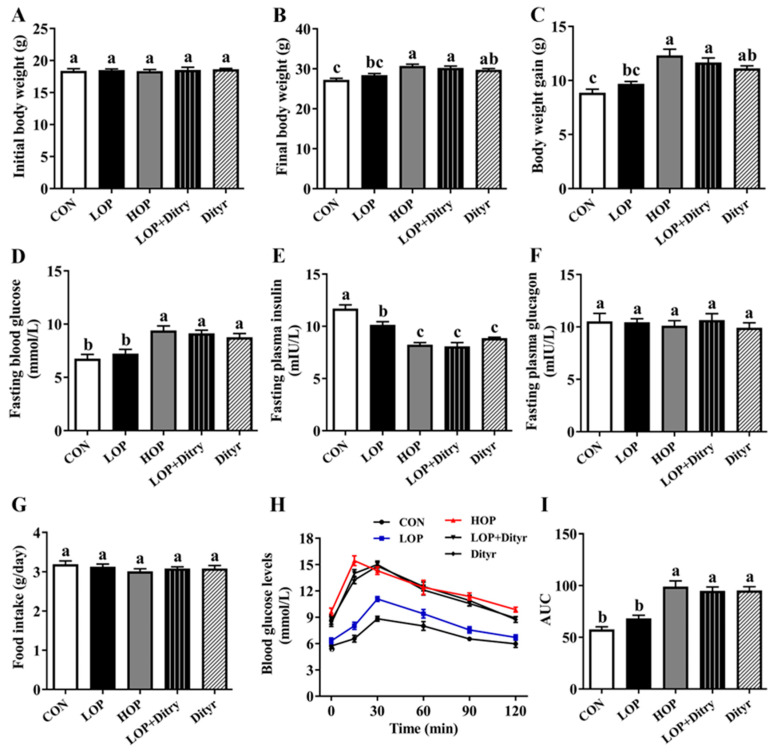
Impacts of Dityr on body weight (BW), food intake, fasting blood glucose, fasting plasma insulin, glucagon levels, and glucose tolerance. (**A**) Initial BW. (**B**) Final BW. (**C**) BW gain. (**D**) Fasting blood glucose levels. (**E**) Fasting plasma insulin concentrations. (**F**) Fasting plasma glucagon levels. (**G**) Food intake. (**H**) Oral glucose tolerance test (OGTT) curves. (**I**) Area under the curve (AUC) for OGTT. Values are mean ± SEM (*n* = 10). Distinct superscript letters indicate statistically significant differences (*p* < 0.05) among groups.

**Figure 2 foods-14-03220-f002:**
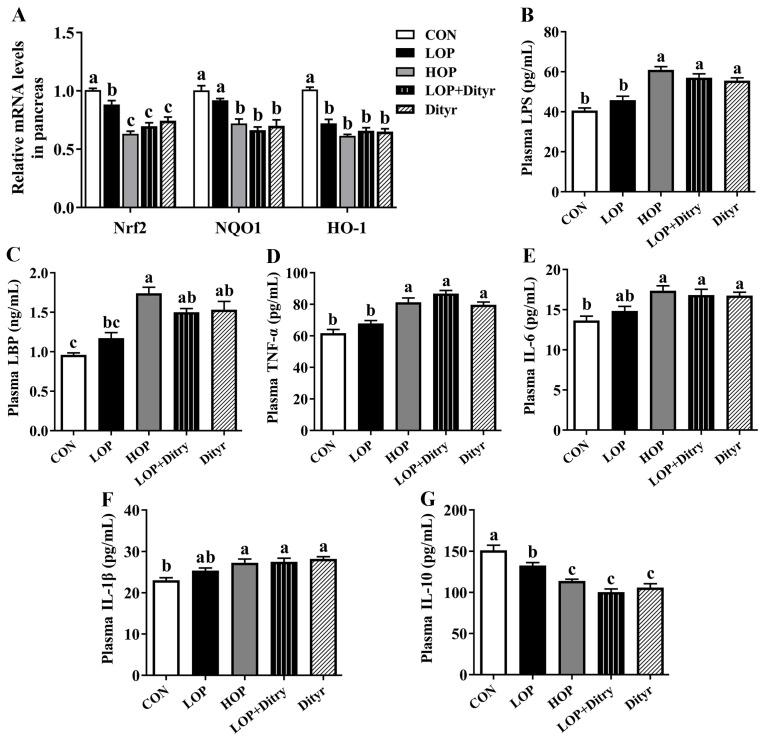
Impacts of Dityr on the expression of pancreatic antioxidant defense-related genes and plasma inflammatory state. (**A**) mRNA expression levels of antioxidant defense-related genes in pancreatic tissue. (**B**) Plasma LPS levels. (**C**) Plasma LBP levels. (**D**–**G**) Plasma inflammatory cytokine levels. Values are mean ± SEM (*n* = 10). Distinct superscript letters indicate statistically significant differences (*p* < 0.05) among groups.

**Figure 3 foods-14-03220-f003:**
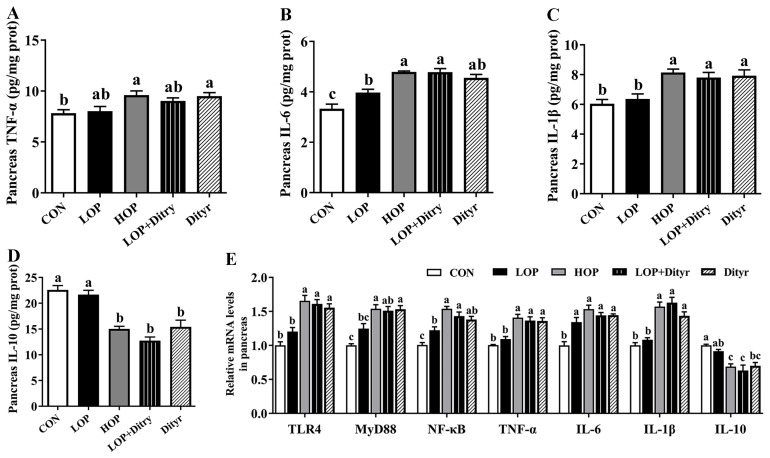
Impacts of Dityr on inflammation state in mouse pancreas. (**A**–**D**) Pancreatic inflammatory cytokine levels. (**E**) mRNA expression of genes associated with inflammatory processes in the pancreas. Values are mean ± SEM (*n* = 10). Distinct superscript letters indicate statistically significant differences (*p* < 0.05) among groups.

**Figure 4 foods-14-03220-f004:**
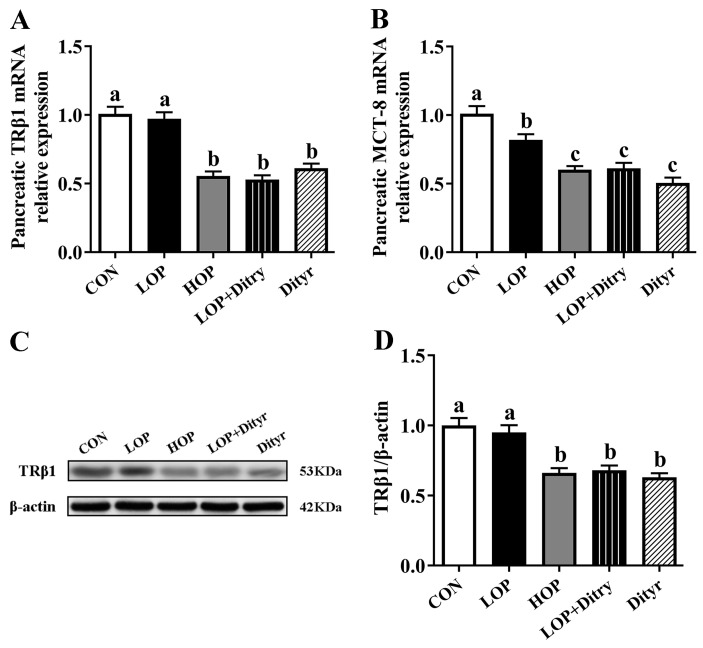
Effects of Dityr on thyroid hormone receptor and transporter in pancreas. Pancreatic (**A**) TRβ1 and (**B**) MCT8 mRNA expression. (**C**,**D**) Protein expression of TRβ1 in pancreas. Values are mean ± SEM (*n* = 10). Distinct superscript letters indicate statistically significant differences (*p* < 0.05) among groups.

**Figure 5 foods-14-03220-f005:**
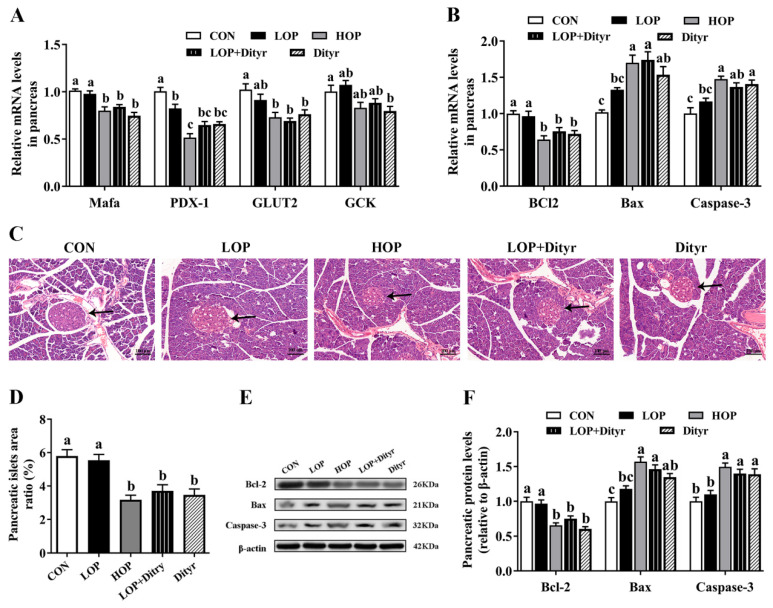
Effects of Dityr on thyroid hormone-regulated β-cell function and apoptosis. (**A**) Pancreatic mRNA expression of insulin secretion regulators. (**B**) Pancreatic β-cell apoptosis-related gene expression. (**C**) H&E-stained pancreatic sections (200×); arrows indicate islets. (**D**) Islet area quantification. (**E**,**F**) β-cell apoptosis-related protein expression. Values are mean ± SEM (*n* = 10). Distinct superscript letters indicate statistically significant differences (*p* < 0.05) among groups.

**Table 1 foods-14-03220-t001:** Impacts of Dityr on oxidative damage in mouse plasma and pancreas.

	CON	LOP	HOP	LOP + Dityr	Dityr
Plasma					
Dityr (pg/mL)	137.71 ± 7.63 ^c^	151.80 ± 7.24 ^c^	202.16 ± 7.43 ^b^	201.23 ± 5.26 ^b^	249.57 ± 9.59 ^a^
AOPPs (pmol/L)	243.64 ± 4.25 ^b^	256.58 ± 7.81 ^b^	298.60 ± 15.40 ^a^	293.95 ± 6.42 ^a^	315.21 ± 7.40 ^a^
MDA (nmol/mL)	7.13 ± 0.27 ^c^	9.96 ± 0.65 ^b^	13.08 ± 0.68 ^a^	12.47 ± 0.57 ^a^	12.47 ± 0.73 ^a^
AGEs (ng/L)	92.92 ± 1.74 ^b^	93.32 ± 3.66 ^b^	106.01 ± 2.84 ^a^	97.04 ± 2.33 ^ab^	98.21 ± 4.06 ^ab^
Pancreas					
Dityr (pg/mg prot)	14.09 ± 0.61 ^b^	17.04 ± 0.64 ^b^	22.09 ± 1.34 ^a^	21.58 ± 0.91 ^a^	23.25 ± 1.11 ^a^
AOPPs (pmol/g prot)	14.73 ± 0.81 ^c^	17.32 ± 0.73 ^bc^	25.86 ± 1.16 ^a^	22.55 ± 1.12 ^a^	21.25 ± 1.06 ^ab^
MDA (nmol/mg prot)	3.07 ± 0.08 ^c^	3.94 ± 0.16 ^bc^	5.30 ± 0.21 ^a^	4.65 ± 0.30 ^ab^	4.98 ± 0.17 ^a^
AGEs (ng/g prot)	13.36 ± 0.60 ^c^	17.92 ± 0.77 ^b^	22.93 ± 0.46 ^a^	20.96 ± 0.88 ^a^	17.99 ± 0.68 ^b^

Note: Values represent mean ± SEM (*n* = 10). Distinct superscript letters indicate statistically significant differences among groups (*p* < 0.05).

**Table 2 foods-14-03220-t002:** Impacts of Dityr on redox state in mouse plasma and pancreas.

	CON	LOP	HOP	LOP + Dityr	Dityr
Plasma					
ROS (RLUs/mL)	688.45 ± 41.52 ^b^	938.27 ± 59.13 ^b^	1318.41 ± 91.63 ^a^	1293.45 ± 51.89 ^a^	1199.60 ± 48.01 ^a^
T-AOC (U/mL)	4.50 ± 0.14 ^a^	4.12 ± 0.11 ^a^	3.58 ± 0.13 ^b^	3.35 ± 0.14 ^b^	3.51 ± 9.13 ^b^
SOD (U/mL)	254.69 ± 10.30 ^a^	244.55 ± 8.09 ^a^	196.99 ± 10.02 ^b^	210.32 ± 11.20 ^ab^	192.66 ± 10.24 ^b^
GSH-Px (U/mL)	134.69 ± 5.21 ^a^	116.55 ± 5.23 ^a^	84.99 ± 3.61 ^b^	80.66 ± 2.91 ^b^	91.66 ± 3.12 ^b^
Pancreas					
ROS (RLUs/mg prot)	943.45 ± 63.07 ^b^	1213.27 ± 71.28 ^b^	1593.41 ± 113.76 ^a^	1568.45 ± 64.69 ^a^	1649.60 ± 78.23 ^a^
T-AOC (U/mg prot)	0.65 ± 0.05 ^a^	0.57 ± 0.02 ^a^	0.34 ± 0.03 ^b^	0.39 ± 0.03 ^b^	0.37 ± 0.03 ^b^
SOD (U/mg prot)	219.01 ± 11.25 ^a^	223.34 ± 8.73 ^a^	169.34 ± 9.37 ^b^	166.67 ± 3.30 ^b^	176.01 ± 8.81 ^b^
GSH-Px (U/mg prot)	154.67 ± 20.87 ^a^	143.67 ± 6.80 ^ab^	100.67 ± 5.11 ^c^	108.67 ± 7.13 ^c^	119.34 ± 5.18 ^bc^

Note: Values represent mean ± SEM (*n* = 10). Distinct superscript letters indicate statistically significant differences among groups (*p* < 0.05).

## Data Availability

The original contributions presented in the study are included in the article/Appendix A, further inquiries can be directed to the corresponding authors.

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
