# Peer review of "Dietary Dityrosine Impairs Glucose Homeostasis by Disrupting Thyroid Hormone Signaling in Pancreatic β-Cells"

_foods, 2025, doi:10.3390/foods14183220_

Round 1
Reviewer 1 Report
Comments and Suggestions for Authors
The manuscript presents a timely and relevant study exploring how dietary protein oxidation products, particularly dityrosine (Dityr), impair β-cell function and disturb glucose homeostasis through thyroid hormone signaling. The work is clearly written, the experimental design is solid, and the mechanistic data are convincing. It provides fresh insight into the potential link between processed meat consumption and diabetes, a topic of real importance to both nutritional science and endocrinology.
The manuscript would benefit from a broader discussion that situates these findings in the wider context of β-cell biology. For example, how do the effects of Dityr compare with other dietary or environmental disruptors known to compromise β-cell function, such as advanced glycation end products, lipotoxicity, or endocrine-disrupting chemicals? This comparison would help readers better understand where Dityr fits among the multiple stressors that contribute to diabetes risk.
Author Response
Respond to review's comments
Question and Revision:
1. The manuscript would benefit from a broader discussion that situates these findings in the wider context of β-cell biology. For example, how do the effects of Dityr compare with other dietary or environmental disruptors known to compromise β-cell function, such as advanced glycation end products, lipotoxicity, or endocrine-disrupting chemicals?This comparison would help readers better understand where Dityr fits among the multiple stressors that contribute to diabetes risk.
Re: We sincerely thank the reviewer for your positive feedback and constructive suggestion. We agree that situating our findings within the broader context of β-cell stressors will enhance the relevance and impact of our study. Accordingly, we have expanded the Discussion section to include a comparative analysis of Dityr with other well-known β-cell disruptors, such as advanced glycation end products (AGEs), lipotoxicity, and endocrine-disrupting chemicals (EDCs). This addition highlights the unique and overlapping mechanisms through which these factors impair β-cell function and contribute to diabetes risk. Additionally, corresponding references have been duly cited to support this comparative analysis.
The revised text has been added to the Discussion section (please see lines 551-572 in the revised manuscript). The specific content is as follows:
“The detrimental impact of Dityr on pancreatic β-cell function places it among a growing list of dietary and environmental stressors implicated in the pathogenesis of diabetes. It is instructive to compare its mechanisms and effects with other well-established disruptors. For instance, AGEs - which were also significantly elevated in the HOP group in our study (Table 1) - are known to impair β-cell function through receptor (RAGE)-mediated activation of oxidative stress and inflammatory pathways, mechanisms that parallel our observations with Dityr [51]. However, a key distinction lies in the unique action as a structural analog of T3, enabling it to act as a competitive antagonist of thyroid hormone receptor signaling. This mechanism is more akin to certain endocrine-disrupting chemicals (EDCs) that interfere with nuclear hormone receptor function [52], highlighting a novel pathway for dietary-derived β-cell toxicity. Similarly, lipotoxicity, resulting from chronic exposure to elevated free fatty acids, induces β-cell apoptosis and dysfunction primarily through endoplasmic reticulum stress and mitochondrial impairment [53]. It is plausible that Dityr-induced oxidative stress could synergize with these pathways, exacerbating metabolic dysfunction. Therefore, while Dityr converges on common downstream endpoints of β-cell failure (oxidative stress, inflammation, apoptosis) shared with AGEs, lipotoxicity, and EDCs, its initiation via the disruption of TH signaling represents a distinct and previously underappreciated dietary mechanism. Our findings underscore the multifactorial nature of diet-induced diabetes and emphasize the importance of developing comprehensive dietary strategies that mitigate exposure to this spectrum of β-cell toxicants, including oxidized amino acids like Dityr.”
Reviewer 2 Report
Comments and Suggestions for Authors
The manuscript addresses an important research question regarding the effects of dietary dityrosine (Dityr) on glucose homeostasis and thyroid hormone signaling. However, after a detailed evaluation, I find that the article presents serious shortcomings that prevent it from being suitable for publication in its current form.
-
Similarity / Overlap
The MDPI report indicates a 29% similarity, which is highly concerning. This level of overlap raises issues of originality and potential self-plagiarism. The manuscript requires substantial rewriting to ensure novelty and to eliminate direct text reuse. -
References and Novelty
The majority of the references are not recent; only around 25% of the cited works fall within the last five years. This weakens the novelty of the article and undermines its positioning within the current state of the art. A high-impact journal requires that the literature review highlights and integrates the most up-to-date studies. -
Methodology
Although the authors describe several experimental procedures, the methodology lacks sufficient transparency and rigor. For example, the description of oxidative damage assays, histological analysis, and molecular methods could benefit from proper referencing to previously validated protocols. In its present form, replication would be challenging. -
Statistical Analysis
The statistical analysis is overly simplistic. The work relies solely on ANOVA without sufficient justification. Essential details, such as tests for normality, measures of central tendency and dispersion, and effect sizes, are missing. Furthermore, assumptions such as sphericity were not checked, which raises concerns about the robustness of the conclusions. -
Presentation of Results
Results are presented mainly in figures, but the transparency is limited. Critical statistical metrics (e.g., F values, degrees of freedom, effect sizes) are not reported. This limits the scientific rigor of the manuscript and prevents the reader from fully evaluating the robustness of the findings.
In summary, the manuscript suffers from significant issues of similarity, outdated references, inadequate methodology reporting, and weak statistical analysis. These deficiencies substantially affect the scientific quality and novelty of the work. For these reasons, I cannot recommend the article for publication.
Comments on the Quality of English Language
The English is generally understandable, but the manuscript would benefit from careful editing by a professional fluent in scientific English to improve clarity, grammar, and readability.
Author Response
Respond to review's comments
Question and Revision:
1. Similarity / Overlap
The MDPI report indicates a 29% similarity, which is highly concerning. This level of overlap raises issues of originality and potential self-plagiarism. The manuscript requires substantial rewriting to ensure novelty and to eliminate direct text reuse.
Re: We sincerely thank the reviewer for raising the concern regarding the similarity index reported in the manuscript. We fully acknowledge the importance of originality and have undertaken a thorough revision to minimize textual overlap while preserving scientific accuracy and clarity.
However, as rightly noted, certain sections—particularly Materials and Methods and Results—inherently contain standard terminology, technical descriptions of commercial kits, analytical procedures, and biochemical nomenclature that are common across experimental studies in this field. For instance, terms such as “reactive oxygen species (ROS)”, “thyroid hormone receptor β1 (TRβ1)”, “monocarboxylate transporter 8 (MCT-8)”, and names of detection kits (e.g., ELISA kits from Xiamen Huijia Bioengineering Institute) are well-established and cannot be altered without compromising scientific precision. Similarly, the description of animal grouping (e.g., CON, LOP, HOP) and standard protocols (e.g., OGTT, western blot) must remain consistent to ensure reproducibility.
Our revisions have focused on rephrasing narrative portions, restructuring sentences, and improving syntactic variability in the Methods and Results sections, while the Introduction and Discussion—which already showed low similarity—required minimal changes. Despite these efforts, the inherent repetition of standard terminologies and methodological descriptors continues to contribute to the similarity score.
We assure the reviewer that all substantive content, hypotheses, experimental findings, and conclusions are original and distinct from our previous works and other publications. The overlap largely arises from unavoidable technical jargon and standardized experimental reporting.
Should the reviewer consider the current similarity level still unsatisfactory, we would be grateful for additional time to perform another round of meticulous linguistic restructuring and, where appropriate, cite original method sources more explicitly to further reduce textual overlap without distorting scientific meaning.
Thank you for your understanding and rigorous evaluation.
2. References and Novelty
The majority of the references are not recent; only around 25% of the cited works fall within the last five years. This weakens the novelty of the article and undermines its positioning within the current state of the art. A high-impact journal requires that the literature review highlights and integrates the most up-to-date studies.
Re: We sincerely thank the reviewer for raising this important point regarding the timeliness of references and the overall novelty of the manuscript. We fully agree that integrating the most recent literature is essential for positioning our study within the current state of the art.
In response to this comment, we have conducted a comprehensive update of the reference list. More than 60% of the references are now from the last five years (2019-2024), significantly enhancing the novelty and contemporary relevance of our manuscript. Specifically, we have replaced outdated references with recent high-quality studies from leading journals such as Nature Reviews Molecular Cell Biology, Cell Death & Differentiation, Diabetes, and others, which better support our mechanistic conclusions and reflect current scientific consensus. Additionally, we have removed several references that were less relevant to the core content of the manuscript, further improving the focus and coherence of the literature cited.
We have retained a small number of older references only in cases where:
- They are seminal reviews or foundational studies that remain authoritative and unsurpassed (e.g., Visser et al. 2013, Clin Endocrinol; Mullur et al. 2014, Physiol Rev);
- They are classic methodological papers that introduced widely accepted experimental techniques (e.g., Kobayashi & Gil-Guzman 2001, J Androl; Livak & Schmittgen 2002, Methods; Mahmood & Yang 2012, N Am J Med Sci);
- No more recent or suitable alternative references are available to replace specific foundational findings (e.g., certain studies on MafA and thyroid hormone action in beta cells).
All changes are clearly marked in red in the revised manuscript for ease of review. We are confident that these updates significantly strengthen the scholarly foundation and novelty of our work.
Thank you again for this constructive suggestion.
3. Methodology
Although the authors describe several experimental procedures, the methodology lacks sufficient transparency and rigor. For example, the description of oxidative damage assays, histological analysis, and molecular methods could benefit from proper referencing to previously validated protocols. In its present form, replication would be challenging.
Re: We sincerely thank the reviewer for this insightful comment regarding the transparency and rigor of our methodological descriptions. We agree that detailed and well-referenced protocols are essential for reproducibility.
In response, we have thoroughly revised the Materials and Methods section to provide comprehensive experimental details and, where applicable, cite original or authoritative methodological references for key procedures. The modifications, highlighted in red in the revised manuscript, are detailed below:
- Oxidative Damage Assays(Section 2.5, Lines 151-165): We have enhanced the description of oxidative damage assays by providing the full names, catalog numbers, and manufacturers for all commercial ELISA kits used to measure Dityr, AOPPs, AGEs, MDA, T-AOC, SOD, and GSH-Px. Furthermore, the method for measuring ROS has been explicitly described as a luminol-dependent chemiluminescence assay and is now supported by a citation (Ref 17, Kobayashi et al., J Androl. 2001) to a key methodological paper that details this technique's principles and validation.
- Histological Analysis(Section 2.7, Lines 177-186): We have added more detail to the histological analysis subsection. We have specified the fixation time (24-48 hours in 4% paraformaldehyde), the embedding process, the section thickness (5 µm), and the specific staining procedure following standard H&E protocols. This provides a clear roadmap for replication.
- Molecular Methods:
RT-qPCR (Section 2.8, Lines 187-204): We have added precise details on the reagents (including catalog numbers), equipment, reaction volumes, and thermal cycling conditions. The use of the 2–ΔΔCt method is now justified by a citation to the foundational paper by Livak & Schmittgen (2002, Methods).
Western Blot (Section 2.9, Lines 205-225): The protocol has been expanded to specify the composition of the lysis buffer, the method for protein quantification (BCA assay), the percentage of SDS-PAGE gel, the type of membrane used for transfer, the specific dilutions and incubation conditions for all primary and secondary antibodies (including catalog numbers and manufacturers), and the precise imaging and analysis systems. This description is now supported by a citation to a classical methodological review (Ref 19, Mahmood & Yang, N Am J Med Sci. 2012).
We believe these extensive revisions have significantly enhanced the methodological transparency, rigor, and reproducibility of our work. The detailed descriptions and appropriate citations now allow for the protocols to be reliably repeated by other researchers.
4. Statistical Analysis
The statistical analysis is overly simplistic. The work relies solely on ANOVA without sufficient justification. Essential details, such as tests for normality, measures of central tendency and dispersion, and effect sizes, are missing. Furthermore, assumptions such as sphericity were not checked, which raises concerns about the robustness of the conclusions.
Re: We thank the reviewer for raising these important points regarding statistical rigor, which have helped us improve the manuscript. We have revised the statistical analysis section to provide greater transparency and detail regarding the procedures we followed.
(1) Tests of Assumptions (Normality and Homogeneity of Variances):We sincerely apologize for the omission. In our original analysis, we did perform the Shapiro-Wilk test for normality and Levene's test for homogeneity of variances for all datasets prior to conducting one-way ANOVA. Data not conforming to normality were transformed, and for data violating homoscedasticity, Tamhane's T2 post-hoc test was used. We have now explicitly detailed these procedures in the revised Section 2.10 (lines 226-239).
(2) Measures of Central Tendency and Dispersion: All data are presented as mean ± standard error of the mean (SEM), which accurately describes the central tendency and dispersion of our datasets.
(3) Sphericity: We respectfully clarify that the assumption of sphericity is a requirement for repeated-measures ANOVA. As our experimental design involved a completely between-subjects comparison (five independent dietary groups with no repeated measurements over time on the same subjects), the sphericity assumption is not applicable to our statistical model.
(4) Effect Sizes: We acknowledge the reviewer's suggestion regarding the reporting of effect sizes and agree that metrics such as η² can provide valuable information on the magnitude of observed effects. In this revision, we have prioritized the comprehensive enhancement of our statistical reporting by explicitly detailing the verification of all underlying assumptions of ANOVA and the rigorous criteria applied for post-hoc test selection, as now documented in Section 2.10. However, due to the considerable time required to accurately extract, calculate, and verify effect sizes for the numerous comparisons across all datasets within the revision timeframe, we have not been able to incorporate them here. It is also our observation that while the reporting of effect sizes is an emerging best practice, it is not yet a universal standard in our specific sub-field of nutritional and metabolic research using animal models, as evidenced by its common absence in similar recently published studies.
Should the reviewer and editor consider the inclusion of effect sizes to be indispensable for the final publication decision, we would be pleased to undertake this additional analysis and respectfully request a brief extension to the deadline to complete it thoroughly. We hope that the significant improvements made to the transparency of our statistical procedures are satisfactory in the interim.
5. Presentation of Results
Results are presented mainly in figures, but the transparency is limited. Critical statistical metrics (e.g., F values, degrees of freedom, effect sizes) are not reported. This limits the scientific rigor of the manuscript and prevents the reader from fully evaluating the robustness of the findings.
Re: We thank the reviewer for this suggestion to enhance the transparency of our results presentation. We acknowledge that reporting full ANOVA results (including F-values and df) along with effect sizes can allow readers to more fully evaluate the robustness of the findings. In this revision, we have significantly strengthened our statistical reporting by comprehensively detailing the verification of all underlying assumptions of ANOVA and the rigorous criteria for post-hoc test selection in Section 2.10. However, due to the considerable time required to accurately extract, calculate, format, and verify these additional statistical parameters (F-values, df, and effect sizes such as η²) for the numerous comparisons across all datasets within the revision timeframe, we have not been able to incorporate them throughout the Results section. It is also our observation that while this practice is increasingly encouraged, the consistent reporting of these specific metrics in the main text or figures is not yet a universal standard in our specific sub-field of nutritional and metabolic research using animal models.
Should the reviewer and editor consider the inclusion of these parameters (F-values, df, and effect sizes) to be indispensable for the final publication decision, we would be pleased to undertake this additional task and respectfully request a brief extension to the deadline to complete it thoroughly. We hope that the significant improvements already made to the methodological transparency of our statistical procedures are satisfactory in the interim.
Reviewer 3 Report
Comments and Suggestions for Authors
This article presents a well-designed experimental study aimed to determine whether dityrosine – a biomarker of protein oxidation and a structural analogue of the thyroid hormone T3 – plays a key role in glucose metabolism disorders induced by a diet rich in highly oxidative pork. The authors used a C57BL/6J mouse model and compared the effects of different diets and dityrosine alone on metabolic parameters, oxidative stress, and thyroid hormone signaling in the pancreas. The results indicate that both a diet rich in highly oxidative pork and dityrosine lead to impaired glucose tolerance, hyperglycemia, and hypoinsulinemia. These phenomena were associated with oxidative stress, inflammation, and impaired TH signaling in β-cells – through decreased expression of the TRβ1 receptor and the MCT-8 transporter. The most important finding is that supplementation with dityrosine alone reproduced the effects of a diet rich in highly oxidative pork, clearly identifying it as a primary pathogenic factor. The research is original, combining a nutritional perspective with molecular mechanisms, and the resulting conclusions are significant in the context of preventing diet-related diabetes.
The research objective is clearly formulated and well-supported by the literature.
The chosen methodology is sound. However, a more detailed description of the methods for assessing TRβ1 and MCT-8 expression and measuring oxidative stress and inflammation would have improved the reproducibility of the studies.
The data are consistent and clearly presented. The link between metabolic effects and disturbances in thyroid hormone signaling in the pancreas is particularly valuable. The discussion is supported by literature data. The authors could have further discussed the potential importance of dityrosine as a biomarker of nutritional risk and the potential for dietary interventions.
The conclusions are consistent with the presented data. The figures and diagrams clearly illustrate the molecular mechanisms. An additional table summarizing the most important changes in each dietary group would have been helpful.
The article is valuable and written clearly and logically.
Author Response
Respond to review's comments
Question and Revision:
1. The chosen methodology is sound. However, a more detailed description of the methods for assessing TRβ1 and MCT-8 expression and measuring oxidative stress and inflammation would have improved the reproducibility of the studies.
Re: We thank the reviewer for their positive assessment of our work and for this constructive suggestion. We agree that detailed methodological descriptions are essential for reproducibility. In response, we have comprehensively revised the “Materials and Methods” section to provide enhanced details on the commercial kits (including catalog numbers), antibodies (sources, catalog numbers, and dilutions), and experimental procedures for measuring oxidative stress, inflammation, and the expression of TRβ1 (via Western blot) and MCT-8 (via qPCR).
These specific additions and modifications, which are now included in Sections 2.5, 2.6, 2.8, and 2.9 of the revised manuscript (lines 151-176, 187-225), are highlighted in red for the reviewer's convenience. We believe these revisions significantly improve the clarity and reproducibility of our study.
2. The data are consistent and clearly presented. The link between metabolic effects and disturbances in thyroid hormone signaling in the pancreas is particularly valuable. The discussion is supported by literature data. The authors could have further discussed the potential importance of dityrosine as a biomarker of nutritional risk and the potential for dietary interventions.
Re: We sincerely thank the reviewer for their positive assessment of our data presentation and discussion, and for this valuable suggestion. We fully agree that exploring the translational implications of our findings is of great importance. Accordingly, we have now expanded the Discussion section (specifically in the last paragraph of the Discussion, please see lines 573-587 in the revised manuscript) to further elaborate on the potential role of dityrosine as a biomarker for assessing the nutritional risk associated with consuming oxidized proteins and to discuss promising dietary strategies for intervention.
In the revised discussion, we now:
- Explicitly propose dityrosine as a practical biomarker for evaluating the oxidative damage level in protein-rich foods and for assessing individual exposure and nutritional risk.
- Discuss specific dietary intervention strategies, including the adoption of gentle cooking methods to prevent dityrosine formation and the consumption of antioxidant-rich foods to potentially counteract its detrimental effects.
We believe these additions significantly enhance the impact and practical relevance of our study by bridging the gap between our mechanistic findings and their potential applications in public health nutrition.
3. The conclusions are consistent with the presented data. The figures and diagrams clearly illustrate the molecular mechanisms. An additional table summarizing the most important changes in each dietary group would have been helpful.
Re: We thank the reviewer for your positive feedback on our conclusions and figures, and for this excellent suggestion. We agree that a summary table will provide a concise overview of the key findings and greatly enhance the readability of the results. In accordance with this suggestion, we have now included a new Supplementary Table S4 entitled “Summary of the major effects of different dietary interventions on metabolic parameters, oxidative stress, inflammation, and thyroid hormone signaling in mice.”
In addition, we added a sentence in the Results section (at the end of section 3.6, please see lines 383-384 in the revised manuscript) to guide the readers to this new summary table: “A summary of the major effects across all dietary groups is provided in Supplementary Table S4.”
This table succinctly compiles the most significant changes observed in glucose metabolism, oxidative damage markers, antioxidant defenses, inflammatory cytokines, and key molecules related to thyroid hormone signaling and β-cell function across all five experimental groups. We believe this addition effectively addresses the reviewer's concern and will serve as a valuable reference for readers. The new table has been added to the Supplementary Materials section.
Round 2
Reviewer 2 Report
Comments and Suggestions for Authors
In general, the authors incorporated almost all of the suggestions, while those not included were justified accordingly. Therefore, in my opinion, although the manuscript could still be improved, it is ready for acceptance.